# Study on the Volatile Organic Compounds and Its Correlation with Water Dynamics of Bigeye Tuna (*Thunnus obesus*) during Cold Storage

**DOI:** 10.3390/molecules24173119

**Published:** 2019-08-28

**Authors:** Xin-Yun Wang, Jing Xie

**Affiliations:** 1Shanghai Engineering Research Center of Aquatic Product Processing & Preservation, Shanghai 201306, China; 2Shanghai Professional Technology Service Platform on Cold Chain Equipment Performance and Energy Saving Evaluation, Shanghai 201306, China; 3National Experimental Teaching Demonstration Center for Food Science and Engineering Shanghai Ocean University, Shanghai 201306, China; 4College of Food Science & Technology, Shanghai Ocean University, Shanghai 201306, China

**Keywords:** bigeye tuna (*Thunnus obesus*), water dynamics, volatile organic compounds, ATP-related compounds, low-field nuclear magnetic resonance, gas chromatography/mass spectrometry (SPME-GC/MS)

## Abstract

Volatile organic compounds (VOCs) and water play a key role in evaluating the quality of aquatic products. Quality deterioration of aquatic products can produce some off-odour volatiles and can induce water content changes. However, no previous study has investigated a correlation between water dynamics and VOCs of bigeye tuna during cold storage. The changes in VOCs, water dynamics and quality attributes of bigeye tuna (*Thunnus obesus*) upon storage at 0 °C and 4 °C for 6 days were investigated. The results showed that the values of ATP, adenosine diphosphate (ADP), adenosine monophosphate (AMP), T_21_ (trapped water) and the relative value of T_1_ decreased (*p* < 0.05), while drip loss and histamine contents increased (*p* < 0.05), which indicated quality deterioration during cold storage. With haematoxylin and eosin (HE) staining, muscle tissue microstructure was observed. VOCs such as hexanal, heptanal, 4-Heptenal, (*Z*)-, pentadecanal-, 1-pentanol, 1-hexanol significantly increased, which sharply increased the content of off-odour volatiles. T_21_ was positively correlated with 1-octen-3-ol, 1-penten-3-ol, while T_21_ was negatively correlated with hexanal, 1-hexanol. Therefore, good correlations between water dynamics and some VOCs were detected during quality deterioration of bigeye tuna throughout cold storage.

## 1. Introduction

Bigeye tuna (*Thunnus obesus*) as an important species dominates the aquatic market due to its high protein and low fat [1]. With the growth of market demand and the achievement of good taste, the quality of bigeye tuna has been paid more attention. Among varieties of preservation techniques, cold storage (−1–8 °C) is the most convenient storage method currently used to delay fresh fish spoilage and prolong its shelf life [2]. According to its high nutritional value and delicious meat, tuna has been investigated for quality changes, including chemical composition, microbial growth, enzymatic activities and physicochemical properties, during cold storage [3].

Water is an important factor for determining the quality of aquatic products. Water accounts for up to 70% of the seafood muscle [4]. Water distribution and mobility in muscle have significant effects on appearance, flavour, ammonia nitrogen content and textural parameters of seafood during storage. High water content can affect quality deterioration of seafood, which leads to a limited shelf-life [5]. Based on the characteristics of sensitivity, non-invasiveness and low cost, LF-NMR (low-field nuclear magnetic resonance) has been applied to aquatic food during processing and storage [4]. Magnetic resonance imaging (MRI) is an accurate and non-destructive method for visualizing the internal aquatic product structure [6]. Accordingly, T_2_ relaxation time has been proven to quantify the water dynamics combining with chemometric models to predict shelf life, thus, has helped to explain how these changes are correlated with fish quality [7].

ATP degradation is used to evaluate the taste or freshness of aquatic products [8]. Aquatic products easily spoil as the initial catabolism of ATP normally results in a fast and temporary accumulation of inosine monophosphate (IMP) [8,9]. IMP and hypoxanthine (Hx) accumulation play an essential role in the flavour of aquatic products, especially for umami or sweet flavours [10]. Volatile organic compounds (VOCs) are formed during the spoilage of aquatic products which is originated from physical, chemical and microbial changes. In addition, VOCs of aquatic products, including aldehydes, ketones, alcohols, aromatic compounds, sulphur compounds, amines, esters and acids can be formed during cold storage, which were analysed by solid phase microextraction coupled with gas chromatography/mass spectrometry (SPME-GC/MS) [11,12,13]. Many researchers have explored that fish spoilage can produce some VOCs, such as 1-pentan-3-ol, 1-octen-3-ol, 1,5-octadien-3-one, (*Z*)-4-heptenal and benzaldehyde [8,14,15]. Hexanal, 2-nonanone and nonanal in yellowfin tuna (*Thunnus albacares*) could be used to evaluate quality changes at 30 °C and in ice using SPME-GC/MS [15]. Therefore, VOCs and ATP-related compounds of fish have been considered important in quality. In addition, VOCs and water are both assessed to quality changes of fish. Water loss can affect the flavour of the fish [4]. However, the affection of different status of water on the specific volatile substance remains unknown. No previous study has been focused on the correlation between water dynamics and VOCs of bigeye tuna throughout cold storage. Therefore, this aspect deserves more attention.

This study aimed to investigate the changes in water distribution and VOCs and their correlation in bigeye tuna to determine the quality during cold storage. Water distribution and mobility were measured by LF-NMR as a rapid and non-destructive method. Changes of physical and chemical property including drip loss, colour, ATP-related compounds and histamine of bigeye tuna during cold storage were also analysed. The study provides novel insights into spoilage mechanisms, that specific VOCs are affected by different status of water.

## 2. Results and Discussion

### 2.1. Physical and Chemical Properties Analysis

Drip loss revealed large variations during cold storage, and the value of drip loss at 4 °C was higher than that of at 0 °C with storage time (Figure 1a). Based on previous findings [16], it was assumed that the drip loss increased due to the myofibrillar fraction denaturation and aggregation.

Colour significantly affected the appearance, storage and acceptability of seafood. The *L** and *a** values significantly decreased with increased storage time (Figure 1b,c). The result indicated that tuna muscle stored for 6 days exhibited a mixed spectrum of metmyoglobin and oxymyoglobin [17]. The decrease in *a** was most probably caused by the outflow of pigment in the exudates that were produced during storage [18]. 

Tuna muscle contains some histidine, which is easy to form histamine. High content of histamine is easy to cause histamine poisoning [19]. Changes in the histamine concentrations of bigeye tuna are shown in Figure 1d. During storage, the histamine concentrations increased. Moreover, histamine concentrations showed more upward trends at 4 °C than that at 0 °C, indicating the effects of temperature on histamine formation. The histamine concentrations increased from 1.5 to 4.50 and 4.77 mg/100 g at 0 °C and 4 °C, respectively, which did not exceed the limit of the 5 mg/100 g (FDA, 1998) standard [20]. The histamine concentration of bigeye tuna increased during cold storage may be caused by the autolysis process catalysed by either histamine oxidase or histamine dehydrogenase [21,22].

### 2.2. ATP-Related Compound Analysis

ATP-related compounds are widely used to evaluate the freshness of fish [8]. From Figure 2a–f, the concentrations of ATP (1.2 μmol/g) and ADP (0.46 μmol/g) of tuna decreased obviously with increased storage time. The concentrations of ATP and ADP degraded remarkably within 2 days. The concentrations of ATP in tuna at 0 °C and 4 °C decreased to 0.2 and 0.18 μmol/g, respectively. Moreover, the level of ATP and IMP in tuna at 4 °C decreased more than at 0 °C. The initial concentrations of IMP (5.07 μmol/g) decreased to 2.53 μmol/g on the 6th day of storage at 0 °C, and to 2.35 μmol/g at 4 °C. However, the concentrations of HxR and Hx appeared to increase. The concentrations of Hx increased slowly with increasing storage time and were still measured at a low concentration at day 6. With the degradation of nucleotides, HxR and Hx were important off-flavour contributors in fish muscle produced during cold storage [10]. The result of tuna obtained in the present study is similar to that reported by Li et al. (2017) [23], who reported that the concentrations of ATP and IMP of white muscle and dark muscle from the common carp (*Cyprinus carpio*) significantly decreased, and the concentrations of HxR and Hx accumulated with storage time increasing as good indicators to evaluate the freshness of the fish. ATP-related compounds were detected to be varied and contributed to the flavour during cold storage. Based on previous study [10], Pongsetkul et al. nucleotides play the role in contributing to the flavour of aquatic products.

### 2.3. Headspace SPME-GC/MS Analysis of VOCs

A total of 29 VOCs, including nine aldehydes, five ketones, nine alcohols and six other compounds, were detected (Table 1). The trend in the VOCs at 4 °C was changed more than that at 0 °C, and various VOCs in the present study were reported previously as products of bacterial metabolism, enzymatic reaction or lipid autoxidation [24,25]. The accumulation of aldehydes, alcohols and ketones were due to the metabolites produced by spoilage microorganisms and fatty acids oxidation [26]. Aldehydes and alcohols are major VOCs as products of chemical activity in fresh seafood. Ketones, acids, aromatic compounds, and other compounds were properly shown in the lower levels (Table 1). Edirisinghe et al. (2007) and Sun et al. (2013) [15,27] made a similar conclusion that hexanal, heptanal, octanal, and nonanal were identified in fresh tuna, and this finding was probably related to derivation from oxidative degradation of polyunsaturated fatty acids [15].

Among aldehydes, hexanal, nonanal, heptanal and propanal were present at the highest concentration on day 0. The high content of hexanal could be closely associated with oxidation of linoleic acid and polyunsaturated fatty acid, which indicated seafood spoilage [28]. Moreover, heptanal, octanal, nonanal, (*E*,*Z*)-2,6-nonadienal and tetradecanal appeared in early storage times, and a fishy odour was confirmed for the above substances [29]. Therefore, the changes in aldehydes can reflect the degree of oxidation of polyunsaturated fatty acids in bigeye tuna, which provides a basis for judging its freshness [8]. 

The compounds from the ketones were relatively low in fresh tuna (day 0). Ketones, acetone and 2,3-pentanedione were only detected on day 0, while 2-heptanone, 2-nonanone and 2-undecanone appeared on day 6 at 4 °C. The abundance of ketones at the end of the storage time was greater than that on day 0, which might have been due to the higher metabolic activity of *Pseudomonads*. The *Pseudomonads* spp, which might increase the ketone concentration, were the most abundant species at the end of storage as shown in an earlier study [29].

Among alcohols, 2-octen-1-ol and (*E*)- and (*Z*)-2-penten-1-ol were not detected in fresh tuna (day 0). The increasing tendency of 1-pentanol, 1-hexanol, 1-octen-3-ol, 1-heptanol and 1-octanol was observed in tuna samples during storage, which showed a similar result, and its increase was previously related to derivation from auto-oxidation of unsaturated fatty acids and arachidonic acid by 12-lipoxygenase [29,30]. Therefore, alcohols are important contributors to off-flavours due to their low odour score [31].

Among aromatic compounds, benzaldehyde were detected because it arose from the degradation of amino acids and because of its pleasant stone fruit and almond nutty aroma [32]. Finally, nonanoic acid, ethyl ester and hexanoic acid increased from the middle stages of storage (day 3) until the end of shelf life (day 6). During seafood spoilage, the accumulation of nonanoic acid, ethyl ester and hexanoic acid have been detected in seafood due to aerobic conditions and high water content accelerating growth and metabolic activity of pseudomonads [29]. The other VOCs detected in bigeye tuna were acids, aromatic compounds and others, which were present at low concentrations. 

### 2.4. LF-NMR Analysis

The water distribution and dynamics were investigated in bigeye tuna during cold storage. From Figure 3a, the CPMG signals decayed slower with increased time, suggesting increased water mobility, and the trend was closely correlated with the different water content in the samples during cold storage. From the multi-exponential fitting of the transverse relaxation data, the T_2_ relaxation spectra of different storage times was obtained (Figure 3b). Three water populations were defined as T_2b_, T_21_ and T_22_, which were present for bound water, trapped water and free water, respectively [33]. Relaxation time, T_2b_, of the bigeye tuna was approximately 0.2–1.74 ms, and the trend was not obvious, which might have been due to water molecules tightly associating with protein [34]. T_21_ was considered to be located in the protein-dense myofibrillar network with a relaxation time of approximately 24–132 ms. Both T_21_ value and the amplitude of T_21_ declined with increasing storage time. Moreover, T_21_ significantly decreased during process, and the variations in 4 °C group was more apparent than that of the 0 °C group. The water molecular mobility increased with the temperature rising. This result was consistent with the previous study that could be explained by destruction of the pores of tuna muscle and water loss, which was due to bacterial growth [7]. The component peak with the longest relaxation time, T_22_, with a relaxation time of 403–1032 ms, was located outside of the myofibrils. With increasing storage time, the T_22_ increased at 0 °C and 4 °C. This phenomenon was observed by the water migration and trapped water moving to free water, suggesting the spoilage of the fish [35]. The quantitative intensity of the MRI images is shown in Figure 3c. The H proton density decreased as cold storage time increased. The higher the temperature was, the more significantly the H proton density decreased, which suggests that high temperature accelerated water loss. This result was consistent with the transverse relaxation curve analysis and drip loss analysis.

### 2.5. Histological Analysis in Bigeye Tuna Muscle Tissue

The microstructure of the transverse section of bigeye tuna muscle tissue stored throughout cold storage is shown in Figure 4a–e. The muscle tissues in Figure 4a are presented at 0 day, when the texture of the muscle tissue was relatively intact and arranged closely and uniformly. A small amount of muscle fibre was loose on day 3 at 0 °C, and the microstructure was more easily broken in myofibrils at 4 °C than 0 °C (Figure 4b,c). At the end of the storage time, the intact muscle fibre structure basically disappeared, and the bare muscle fibre space was obvious, with significant cavities and water loss (Figure 4d,e). The distinct gaps on the surfaces of perch muscle were also attributable to internal protein structure degeneration and the external factor of temperature changes [36,37].

### 2.6. Relationship between LF-NMR Parameters (T_2b_, T_21_ and T_22_) and VOCs

The relationship between LF-NMR parameters and VOCs was investigated by RDA (Figure 5) and Pearson correlation coefficients (Table 2). RDA showed that T_2b_ was significantly correlated with acetone, 2-undecanone, 1-hexanol and 1-nonanol (*p* < 0.05). Pearson correlation coefficient results showed extremely significant correlations between T_21_ and hexanal (r = −0.80), 1-hexanol (r = −0.71), 1-octen−3-ol (r = 0.88) and 1-penten-3-ol (r = 0.82). T_22_ was positively correlated with nonanoic acid, ethyl ester (*p* < 0.05), while T_22_ was negatively correlated with naphthalene (*p* < 0.05). With the extension of storage time, T_21_ decreased significantly while T_22_ increased, which resulted in an increase in the metabolic rate and microbial growth. The myofibrils tissue became loose and intramuscular connective of the skelemin damaged due to the value of free water increased during storage, so that microorganisms grew quickly to produce numerous VOCs including ethyl ester, 1-hexanol, nonanoic acid and 1-nonanol [4,38], indicating that accelerated quality deterioration [7]. Therefore, the results showed that water dynamics were significantly correlated with the volatile compounds, which could be applied to evaluate the quality of bigeye tuna.

## 3. Materials and Methods

### 3.1. Sample Preparation

The 3 kg of frozen bigeye tuna from back muscles were purchased from Zhejiang Fenghui Ocean Fishing Company Ltd. (Zhejiang, China) and shipped to the laboratory of Shanghai Ocean University with dry ice. After thawing for approximately 24 h at 0 °C, the fillets were divided into pieces of about 200 g, placed into bags (polyethylene, PE). The samples were labelled and covered with PE, then stored in a refrigerator at 0 °C and 4 °C (KB400, BINDER, Tuttlingen, Germany), respectively. The quality of samples was analysed every day while the VOCs were determined on day 0, day 3 and day 6.

### 3.2. Physical and Chemical Property Determination

#### 3.2.1. Drip Loss and Colour Measurement Analysis

The measurement of drip loss followed the procedure of Kaale et al. [20] with some modifications. The total weight of the package, fish and leaking juices were determined (*W_1_*), then, the package was cut, the juices were poured out; and the moisture on package and fish was absorbed. The total weight of the package and fish (*W_2_*) and the weight of package (*W_3_*) were determined. The drip loss (*W*) was calculated according to the following formula,
$$w=\frac{W_{1}-W_{2}}{W_{1}-W_{3}}\times 100\%$$

The colour measurement of bigeye tuna was performed with a digital Konica colorimeter CR-400 (Tokyo, Japan). The *L** (lightness) and *a** (redness and greenness) values were recorded, and the mean value was calculated (*n* = 6). The instrument was calibrated using a white standard before measuring [39].

#### 3.2.2. Histamine Analysis

Histamine analysis was based on the method of Köse & Hall [40]. Tuna sample (2 g) was homogenized with 10 mL of 0.4 mol/L trichloroacetic acid (TCA) solution and then centrifuged for 15 min at 3000 r/min. The supernatant was obtained, and the centrifugation was combined twice. The combined supernatants were filtered on paper into a 25 mL volumetric flask with the 0.4 mol/L TCA solution.

A volume of 250 μL of solution was transferred into a 2 mL test tube, and 25 μL of 2 mol/L NaOH was used to adjust the solution to pH 13 in order to make a strongly basic environment for the derivatization reaction. Then, 75 μL of the saturated Na_2_CO_3_ solution and 500 μL of 10 mg/mL dansyl chloride was added to the solution in the dark at 40 °C for 45 min. A volume of 25 μL of 25% concentrated ammonia in water was added to the solution to stand for 30 min, and then 375 µL of acetonitrile was added to the mix. The final solution was filtered through a nominal 0.45 μm membrane filter (ANPLE Laboratory Technologies Co., Ltd., Shanghai, China) for standby.

The HPLC (LC-2010C, Shimadzu, Japan) consisted of a Model 2 LC-10ADvp pump, a Model SIL-10ADvp automatic loading sample injector, and a Model SPD-10A(V)vp detector. The HPLC method condition included column: InertSustain C_18_ (4.6 mm × 150 mm, 5 μm), sample injection volume: 20 μL, detection wavelength: 254 nm, isocratic mobile phase: acetonitrile and water (40:60, *v*/*v*), and flow rate: 1 mL/min. The histamine content of bigeye tuna was determined by the HPLC external calibration method was according to the method of Chen et al. [41]. The limit of detection (LOD) was 1 mg/100 g. Relative Standard Deviation (RSD) and stability were 2.68% and 95.8%, respectively. Recoveries of histamine spiked in bigeye tuna sample ranged from 99.8% to 101.0%.

### 3.3. ATP-Related Compounds Analysis

The ATP-related compound was conducted according to Yu et al. with some modifications [8]. The ATP-related compounds were determined using the HPLC (LC-2010C, Shimadzu, Japan). The condition included mobile phase: 0.05 mol/L phosphate, column: InertSustain C_18_ (4.6 mm × 150 mm, 5 μm), detector: SPD-10A(V)vp, flow rate: 1.0 mL/min, and temperature: 30 °C. The standards were detected including adenosine-5′-triphosphate (ATP), adenosine-5′-diphosphate (ADP), adenosine-5′-monophosphate (AMP), inosine-5′-monophosphate (IMP), hypoxanthine (Hx) and inosine (HxR) (Sigma-Aldrich Co., Shanghai, China), separately. Then, ATP-related compounds were detected for calculation of each compound based on the peak areas using external standard.

### 3.4. VOCs Determination by SPME-GC/MS Analysis

Referring to the method of Li et al. [38], 4 g of the minced sample were transferred into 20 mL glass headspace vials and incubated at 40 °C for 15 min to equilibrate the system. The SPME fibre (DVB/CAR/PDMS 50/30 mm) was exposed to headspace and constant length of the fibre for an additional 30 min.

VOCs in bigeye tuna samples were measured using an Agilent 7890A gas chromatograph coupled to an Agilent 5975C mass spectrometer and were separated by a capillary DB-WAX column (30 m × 0.25 mm × 0.25 µm, Agilent, California, USA). The carrier gas was helium (high purity 99.99%). Injector temperature was 260 °C. The temperature of the GC oven was first kept at 40 °C for 5 min with a flow rate of 1.0 mL/min, increased 5 °C/min to 120 °C, then increased 10 °C/min to 250 °C and maintained for 5 min. The condition included the interface temperature: 260 °C, electron energy: 70 eV, a full scan range: 20–400 *m*/*z*, MS source and quadrupole: 230 and 150 °C. Identities of VOCs was used by matching mass spectra or retention time with the National Institute of Standards and Technology (NIST) 14 spectral database and semi-quantitative analysis using the area normalization method of each compound.

### 3.5. LF-NMR

The LF-NMR measurement was carried out according to the method by Wang et al. [7]. The LF-NMR instrument measured at 32 °C and 20 MHz for the 1H NMR Analyser (Meso MR23-060H-I, Niumag Electric Corporation, Shanghai, China). The diameter of the radio frequency coil was 70 mm. The spin-lattice relaxation time, T_1_, was performed by Inversion-Recovery (IR) sequences. The transverse relaxation time, T_2,_ was measured with the Carr–Purcell–Meiboom–Gill (CPMG) sequences to collect decay signals. Accurately weighed bigeye tuna samples covered in commercial plastic food film to avoid water loss inside the tube. The signal inversion data, multiexponential decay curve and normalize the data were obtained by NMRAS analysis software (NMI20-030H-1 NMR analyser: Suzhou Niumag Analytical Instrument Co., Suzhou, China).

### 3.6. Haematoxylin and Eosin (HE) Staining of Bigeye Tuna Muscle

Muscle tissues from bigeye tuna samples were collected and immersed in 10% neutral buffered formalin, dehydrated through a serial alcohol gradient and embedded in paraffin wax. Sections were stained with haematoxylin and eosin (HE) according to Liu et al. [42].

### 3.7. Statistical Analysis

Duncan’s test was used to test for differences between means corresponding to physical and chemical property determination, ATP-related compounds, VOCs and LF-NMR analysis. The correlation coefficient (Pearson’s) and the significance were defined at *p* < 0.05 using SPSS 19.0 software (Chicago, IL, USA), and curves were plotted in Origin 8.6 (OriginLab Corp., Northampton, MA, USA). Redundancy analysis (RDA) used Canoco software version 5.0 to find the correlation between volatile organic compounds and water dynamics. All analyses were conducted with three replicates.

## 4. Conclusions

Changes in the physicochemical properties, ATP-related compounds, water dynamics and VOCs of bigeye tuna during cold storage were investigated. With the extension of storage time, the drip loss and histamine contents were continuously enhanced, *L** and *a** declined significantly and the ATP-related compounds degraded. The microstructure analysis of bigeye tuna samples visually displayed the gradual increase in cell space and the breakage of muscle tissues. Based on the GC-MS analysis and LF-NMR analysis, the VOCs of bigeye tuna mainly consisted of aldehydes and alcohols increased, while the values of T_21_ significantly decreased (*p* < 0.05) during cold storage, indicating that the tuna’s quality, freshness and acceptability decreased quickly. A strong correlation was obtained between the VOCs and T_2_ transverse relaxation time by Pearson correlation analysis and RDA. Therefore, VOCs and its correlation with water dynamics provide unique insights into monitoring the quality changes of bigeye tuna during cold storage.

## Figures and Tables

**Figure 1 molecules-24-03119-f001:**
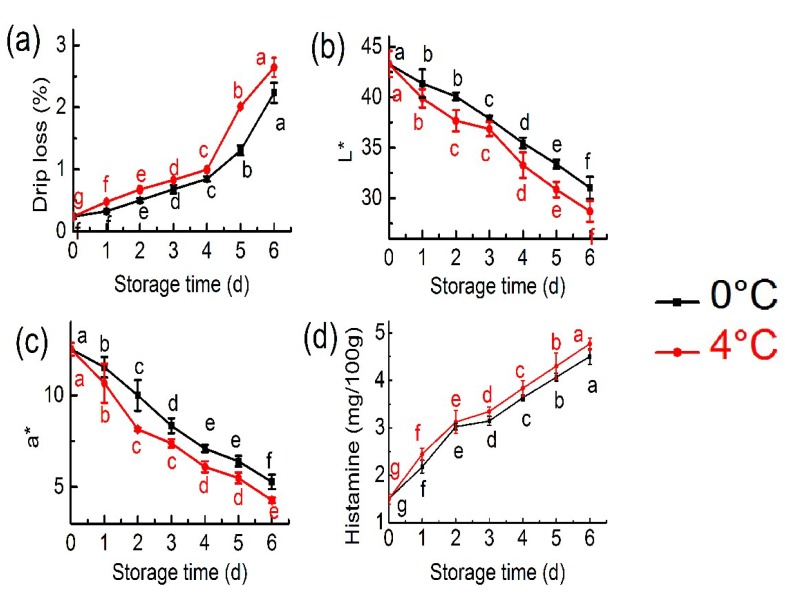
Changes in drip loss (**a**), *L** (**b**), *a** (**c**) and histamine concentrations (**d**) at 0 °C and 4 °C for 6 day (each point is the mean value of three determinations). ^a,b,c,d,e,f,g^ Means in the same column with different superscripts are significantly different (*p* < 0.05).

**Figure 2 molecules-24-03119-f002:**
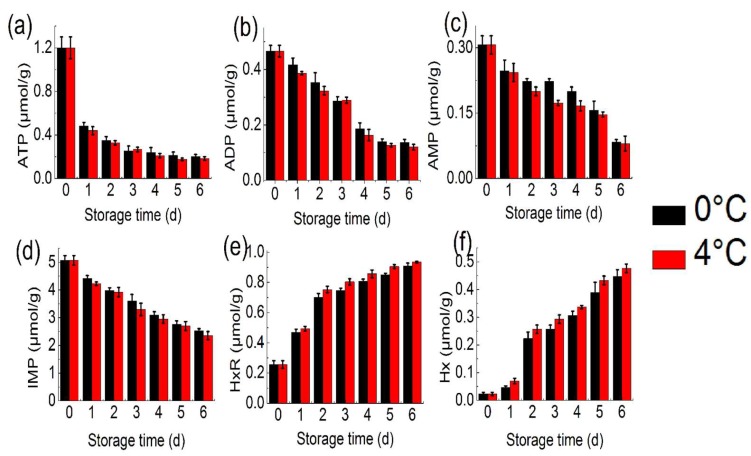
ATP (**a**), adenosine diphosphate (ADP) (**b**), adenosine monophosphate (AMP) (**c**), inosine monophosphate (IMP) (**d**), inosine (HxR) (**e**) and hypoxanthine (Hx) (**f**) content of bigeye tuna at 0 °C and 4 °C for 6 day (each point is the mean value of three determinations)**.**

**Figure 3 molecules-24-03119-f003:**
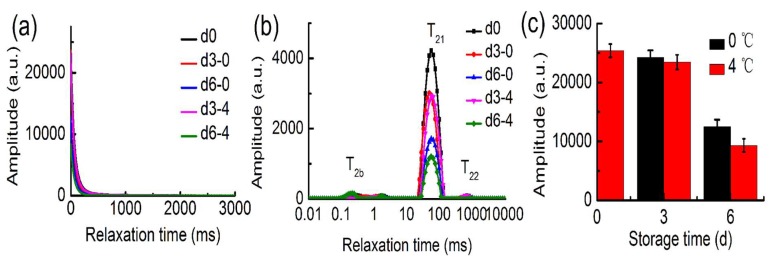
Changes in Carr–Purcell–Meiboom–Gill (CPMG) relaxation decay curves (**a**), transverse relaxation time (**b**) and corresponding relative intensity of the T_1_ weighted images (**c**) for bigeye tuna at 0 °C and 4 °C.

**Figure 4 molecules-24-03119-f004:**
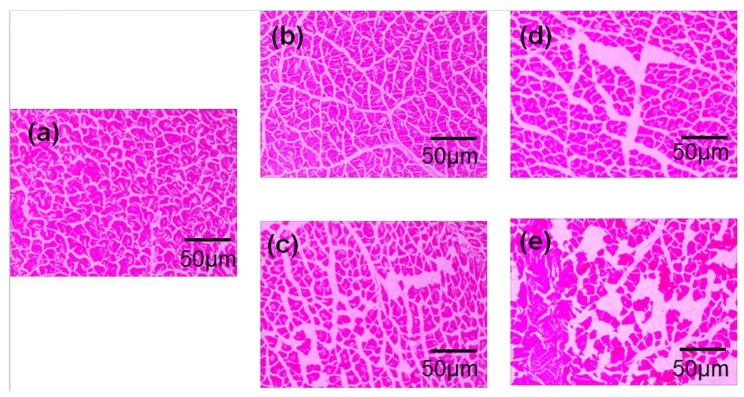
Changes in haematoxylin and eosin (HE)-stained muscle tissues of bigeye tuna stored for 0 days (**a**); tuna stored at 0 °C (**b**,**c**) for 3 days and 4 °C (**d**,**e**) for 6 days.

**Figure 5 molecules-24-03119-f005:**
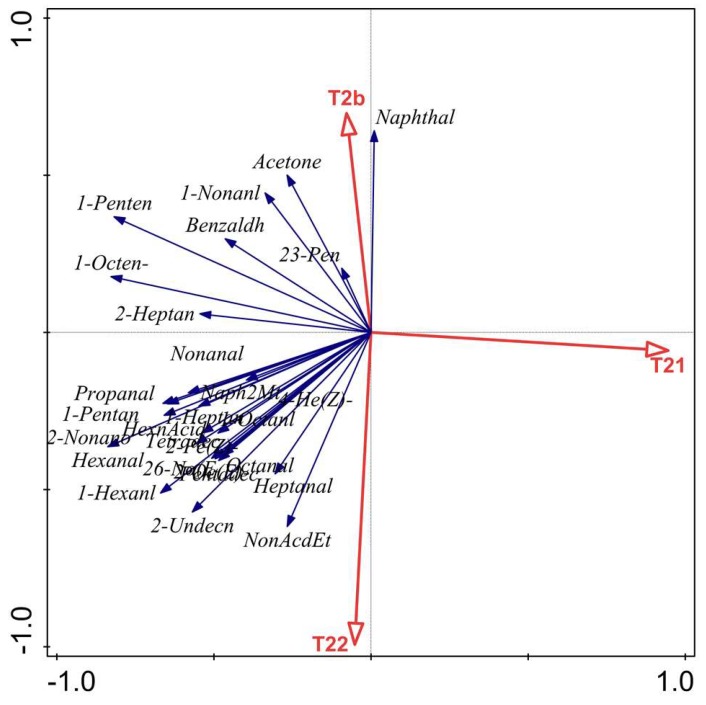
Redundancy analysis (RDA) of T_2_ transverse relaxation and VOCs.

**Table 1 molecules-24-03119-t001:** Main volatile organic compounds (VOCs) and their relative contents (chromatographic peak area ×10^−6^) in tuna at 0 °C and 4 °C for 0, 3, 6 days. Day 3–0 and day 3–4: tuna stored for 3 days at 0 °C and 4 °C, respectively. Day 6–0 and day 6–4: tuna stored for 6 days at 0 °C and 4 °C, respectively. (each point is the mean value of three determinations).

VOCs	Relative Concentration (Area 10^−6^)				
	Day 0	Day 3–0	Day 6–0	Day 3–4	Day 6–4
Aldehydes					
Hexanal	8.61 ± 0.12 ^d^	10.63 ± 0.64 ^d^	31.39 ± 0.69 ^b^	23.72 ± 0.02 ^c^	35.56 ± 2.5 ^a^
Heptanal	5.74 ± 0.12 ^b^	1.12 ± 0.05 ^c^	0.51 ± 0.14 ^c,d^	0.31 ± 0.00 ^c,d^	28.87 ± 0.38 ^a^
Octanal	1.35 ± 0.01 ^b,c^	3.83 ± 0.01 ^b^	2.33 ± 0.06 ^b,c^	0.02 ± 0.00 ^c,d^	18.04 ± 0.27 ^a^
Nonanal	1.21 ± 0.04 ^c^	9.98 ± 0.10 ^b^	1.08 ± 0.03 ^c^	0.02 ± 0.00 ^d^	17.62 ± 0.31 ^a^
2,6-Nonadienal,(*E*,*Z*)-	-	0.01 ± 0.00 ^b^	-	0.02 ± 0.00 ^b^	6.54 ± 0.04 ^a^
Tetradecanal	-	0.01 ± 0.00 ^b^	0.01 ± 0.00 ^b^	0.02 ± 0.00 ^b^	2.95 ± 0.05 ^a^
Propanal	0.79 ± 0.01 ^e^	10.05 ± 0.12 ^b^	4.96 ± 0.14 ^c^	2.83 ± 0.02 ^d^	20.90 ± 0.98 ^a^
4-Heptenal, (*Z*)-	-	0.29 ± 0.01 ^b^	0.28 ± 0.01 ^b^	0.02 ± 0.00 ^c^	5.67 ± 0.05 ^a^
Pentadecanal-	-	0.01 ± 0.00 ^c^	0.50 ± 0.02 ^b^	0.02 ± 0.00 ^c^	16.18 ± 0.20 ^a^
**ketones**					
Acetone	0.04 ± 0.01 ^c^	0.29 ± 0.01 ^b^	0.58 ± 0.01 ^b^	5.23 ± 0.36 ^a^	0.13 ± 0.00 ^b^
2-Heptanone	-	0.05 ± 0.00 ^b^	-	0.08 ± 0.06 ^b^	0.14 ± 0.06 ^a^
2-Nonanone	-	0.04 ± 0.01 ^b^	0.06 ± 0.01 ^b^	0.09 ± 0.03 ^b^	0.76 ± 0.04 ^a^
2-Undecanone	-	-	0.93 ± 0.26 ^a^	-	0.56 ± 0.06 ^b^
2,3-Pentanedione	0.05 ± 0.00 ^a^	0.04 ± 0.00 ^a^	0.08 ± 0.01 ^a^	0.09 ± 0.00 ^a^	0.02 ± 0.00 ^a^
**Alcohols**					
1-Pentanol	0.96 ± 0.18 ^d^	11.53 ± 0.03 ^c^	10.30 ± 0.09 ^c^	23.37 ± 0.00 ^b^	69.16 ± 2.70 ^a^
1-Hexanol	4.03 ± 1.51 ^e^	20.74 ± 0.05 ^c^	83.40 ± 0.93 ^a^	10.56 ± 0.04 ^d^	69.68 ± 50.97 ^b^
1-Octen-3-ol	7.37 ± 2.15 ^d^	77.27 ± 0.27 ^b^	88.0 ± 0.93 ^a^	61.41 ± 0.04 ^c^	65.08 ± 16.45 ^c^
1-Heptanol	1.99 ± 0.74 ^e^	14.20 ± 0.04 ^d^	23.48 ± 0.25 ^c^	56.02 ± 0.03 ^b^	142.15 ± 4.87 ^a^
1-Octanol	7.88 ± 5.21 ^c^	5.55 ± 0.05 ^e^	6.24 ± 0.05 ^d^	12.88 ± 0.04 ^b^	39.19 ± 4.41 ^a^
2-Octen-1-ol, (*E*)-	-	3.08 ± 0.02 ^b^	3.90 ± 0.03 ^b^	0.04 ± 0.00 ^c^	47.04 ± 3.52 ^a^
1-Nonanol	0.82 ± 0.30 ^d^	2.52 ± 0.01 ^c^	3.36 ± 0.03 ^c^	76.97 ± 7.44 ^a^	16.09 ± 0.99 ^b^
2-Penten-1-ol, (*Z*)-	-	0.72 ± 0.01 ^c^	1.74 ± 0.03 ^c^	9.55 ± 0.00 ^b^	55.07 ± 5.35 ^a^
1-Penten-3-ol	10.93 ± 4.95 ^d^	161.64 ± 0.39 ^b^	120.12 ± 1.19 ^c^	129.63 ± 0.20 ^c^	138.87 ± 2.84 ^a^
**Others**					
Nonanoic acid, ethyl ester	0.90 ± 0.79 ^b^	0.02 ± 0.00 ^d^	0.75 ± 0.13 ^c^	0.02 ± 0.00 ^d^	2.63 ± 0.22 ^a^
Hexanoic acid	-	1.64 ± 0.28 ^c^	1.76 ± 0.03 ^c^	2.71 ± 0.04 ^b^	9.73 ± 0.00 ^a^
Benzaldehyde	0.05 ± 0.02 ^b^	0.23 ± 0.06 ^a^	0.06 ± 0.01 ^b^	0.10 ± 0.04 ^a^	0.21 ± 0.03 ^a^
Naphthalene	0.05 ± 0.00 ^b^	0.13 ± 0.02 ^a^	0.08 ± 0.02 ^a,b^	0.10 ± 0.08 ^a^	0.02 ± 0.00 ^b^
Naphthalene,2 methyl-	0.03 ± 0.01 ^b^	0.04 ± 0.01 ^b^	0.02 ± 0.00b ^b^	0.20 ± 0.00 ^b^	0.57 ± 0.25 ^a^

^a, b, c, d, e^ Means in the same row with different superscripts are significantly different (*p* < 0.05).

**Table 2 molecules-24-03119-t002:** Pearson’s correlation analysis and levels of significance for correlations between LF-NMR parameters (T_2b_, T_21_ and T_22_) and VOCs of bigeye tuna.

Parameter	T_2b_	T_21_	T_22_
Hexanal	−0.25	−0.80 **	0.39
Heptanal	−0.18	−0.21	0.48
Octanal	−0.17	−0.40	0.41
Nonanal	−0.02	−0.34	0.18
2,6-Nonadienal,(*E*,*Z*)-	−0.14	−0.38	0.43
Tetradecanal	−0.14	−0.38	0.43
Propanal	−0.06	−0.52 *	0.23
4-Heptenal, (*Z*)-	−0.16	−0.40	0.44
Pentadecanal-	−0.16	−0.40	0.44
Acetone	0.53 *	−0.21	−0.46
2-Heptanone	0.22	−0.46	−0.01
2-Nonanone	−0.11	−0.47	0.39
2-Undecanone	−0.60 *	−0.62 *	0.56 *
2,3-Pentanedione	0.05	−0.15	−0.21
1-Pentanol	0.02	−0.54 *	0.28
1-Hexanol	−0.54 *	−0.71 **	0.51
1-Octen−3-ol	0.01	0.88 **	−0.16
1-Heptanol	0.03	−0.56 *	0.28
1-Octanol	−0.04	−0.38	0.37
2-Octen-1-ol, (E)-	−0.17	−0.42	0.43
1-Nonanol	0.55 *	−0.25	−0.39
2-Penten-1-ol, (Z)-	−0.05	−0.43	0.36
1-Penten-3-ol	0.27	−0.82 **	−0.33
Nonanoic acid, ethyl ester	−0.40	−0.21	0.63 *
Hexanoic acid	−0.03	−0.56 *	0.31
Benzaldehyde	0.29	0.44	−0.27
Naphthalene	0.37	−0.06	−0.65 **
Naphthalene,2 methyl-	0.05	0.43	0.28

Significance was identified as * *p* < 0.05 and ** *p* < 0.01.

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
