# Peer review of "Study on the Volatile Organic Compounds and Its Correlation with Water Dynamics of Bigeye Tuna (Thunnus obesus) during Cold Storage"

_molecules, 2019, doi:10.3390/molecules24173119_

Round 1
Reviewer 1 Report
1. Why did you study the transformation during cold storage at 00C and 40C as long as the preservation was usually done by cold storage at -10C-80C.
2. delete (2004) from line 74.
3. replace ; with , in line 75-76.
4. Describe briefly the HPLC method used for histamine determination. Also, the method used for the quatitation of histamine must be specified.
5. delete (2018) from line 95.
6. What is the carrier gas used for GC determination?
7. delete (2017) from line 109.
8. In the caption of figure 2 the explanation of a-f must be write.
9. The conclusions is not properly write and not clearly underlined the usefulness of the study.
Reviewer 2 Report
The manuscript ‘’Study on the Volatile Organic Compounds and its Correlation with Water Dynamics of Bigeye Tuna (Thunnus obesus) during Cold Storage’’ is interesting and fits into the scope of this journal. In addition, is well written and useful data is presented. I just miss some methodology that needs to be included in the manuscripts in order to allow other researchers to reproduce the experiment.
I recommend publish the manuscript after minor revision.
Specific comments-
Line 45- This statement needs to be supported by at least a reference.
Line 55- ‘’We don’t know…’’- who? Please change the sentence.
Lines 69- Did you analyze the bag as control? Some PE bags may contain some of the compounds you are analyzing, for example- histamine maybe be used as a plastic additive.
Line 71-‘’Identified’’- You mean ‘’determined’’ I guess.
Lines 84-86- The HPLC method needs to be described here, so other researches can reproduce the experiment. What type detector did you use? Was the method validated? What was the LOD? RSD? Recovery? Stability? These parameters must be included in the paper to give confidence to the study. This should be included for the GC-MS analysis as well.
Lines 95-107- Please include where were the analytical standards purchased from.
Lines 123-Duncan test- Please describe in the test.
Why did the authors only analyze histamine and no other biogenic amines such as cadaverine?
What HxR stands for?
Figure 2- Please include in the text what a,b,c…stands for.
Table1- d3-4 and d6-4- What is this? Please explain in detail in the text.
After these minor revision, in my opinion, the manuscript can be accepted for publication.
Reviewer 3 Report
Respectable,
It is in my opinion that the manuscript molecules-580128 entitled "Study on the Volatile Organic Compounds and its Correlation with Water Dynamics of Bigeye Tuna (Thunnus obesus) during Cold Storage" is not suitable for publication in Molecules in its current form but only after a major revision.
The work has many positive components since it deals with an important topic which is quality deterioration during the storage of Bigeye tuna, which is an economically important agricultural product in many countries. The work in general is scientifically sound, the experimental protocol is well executed and the technical procedures were correctly conducted, although with certain ambiguities that need to be clarified. The results could be considered original, novel, interesting, and useful for researchers in the field and beyond, and represent a significant contribution to the field.
The major drawback of the manuscript is the level of writing at some points which currently is not at the level of a high impact journal such as Molecules. In the Abstract and the Introduction, authors should state the problem better and introduce a reader into the topic more clearly and concisely, yet again in simple words. For example, from the text in the Abstract and the Introduction it is not clear (for the readers not so familiar with the specific topic and the corresponding details) WHY the level of water is important and HOW water is connected with quality (more water - lower quality?). Authors should emphasize more the most important findings of the study in the abstract. English should also be significantly improved.
Please find the specific comments that corroborate my opinion:
Lines 13-24: The results and the conclusions in the abstract are too general, one cannot conclude anything about the relationship of VOCs, water, ATP etc. with quality.
Lines 14-15: Why correlation of water and VOCs is important? Please explain in few words.
Line 22: “significantly correlated” – all of them? Positively or negatively? Does all the investigated VOCs have the same meaning and importance (for example, are some of them indicators of microbial spoilage or oxidation, or similar)?
Line 37: “Water is an important factor…” – in which way? It is not clear WHY the level of water is important and HOW water is connected with quality (more water – lower quality?). This issue is much clearer after reading the whole manuscript but it should be clear after reading the Introduction section.
Lines 40-42: Authors state (cite) the source (6) in which was explained how “T2 has proven to quantify the water dynamics and explained how these changes are correlated with fish quality.” But we (the readers) still do not know how, please explain in short.
Line 43: “ATP degradation…” - please add a reference.
Lines 46-53: The construction of these sentences is a little bit odd, please rephrase. For example, line 48: “…during cold storage,…” – what? Were formed / liberated? Please update this sentence.
Line 49: “Many researchers…” but only a single reference – please add more references.
Lines 46-57: More references are needed in general about the relationship of various parameters, especially VOCs and water, with fish quality. For example, a sentence should be added about the origins of volatiles (microbiological, enzymatic, chemical, etc.), which processes occur during storage and produce VOCs, and, if it is possible, how it is connected with the quantity of water. As well, it is not clear if the VOCs are considered only as indicators of the changes during storage, or they actually have sensory significance and (negatively?) affect fish organoleptic quality. Please indicate this, in short.
Lines 82-83: What is a white standard?
Lines 84-86: The authors cited a published paper with details on histamine analysis, and it is therefore not necessary to re-report them all in this paper, however more information is certainly needed. For example, the details about the type of detector (e.g. including the wavelengths) and column, mobile phase composition and type of identification / quantification are necessary. Please update this part.
Lines 87-93: More details are given here but some important details are needed (detector, column, type of quantification (calibration curve, internal standard, etc.)).
Lines 95-96: It is not clear how 72 g in total are related to 4 g pieces subjected for single analyses? Were there any replicated procedures in sampling/analysis? How did authors ensure that the samples used in the analysis (4 g samples) were representative of the whole sample? Were 72 g minced and 4 g pieces were sampled from the whole mass? These details are very important, please indicate all the details.
Lines 99-107: Injector temperature?
Lines 106-107: How was the peak area normalised? Please indicate this. Did author use an internal standard (there is no information)? In the case authors did not normalise peaks the procedure should be referred to as semi-quantification.
Lines 106-107: Were there any replicate analyses? In line 127 authors stated that all analyses were performed in triplicates. It is not clear if standard deviations in Table 1 were calculated form three analyses of the same sample (the same 4 grams) or 4 grams were sampled in triplicates and analysed once each.
Lines 122-127: One-way ANOVA with a post-hoc test is not always the best way to reveal significant differences. For example, please correct me if I am wrong, in Fig 1. 14 points (treatments) were compared and it is not very clear if time or temperature have had a larger effect at each point. It would be interesting to include the results of two-way ANOVA, with storage temperature (0 and 4 degrees C) and duration (0, 3, and 6 days) as factors: in this way better conclusions about the relative magnitudes of their influence could be made. I suggest putting such a table in supplementary files.
Lines 131-132: “The drip loss increased due…” - please use phrases like “It was assumed/is probable that the drip loss increased…as shown in a previous study” or “Based on previous findings (19), it was assumed that the drip loss increased…”. You cannot be 100% sure that myofibrillar fraction denaturation and aggregation was the reason if you did not measure it directly. The same applies for line 136: “The decrease in a* was most probably caused by…”. Please follow this principle thorough the text.
Lines 139-141: The authors compare the dynamics of histamine concentrations at 0 C and 4 C, but, judging from Figure 1d, no significant differences were noted between the 2 temperatures at any of the points of measurement. Again, I am sure 2-way ANOVA would provide different perspective in this case also.
Lines 138-144: Authors should report briefly the significance of histamine as an allergen.
Lines 150-151: Again, authors compare ATP and IMP at 0 C and 4 C, and even use the phrase “dramatically”, but no statistical differences were observed. I suggest 2-way ANOVA.
Lines 145-159: Authors should state what these results mean practically. For example, more discussion is needed about the contribution of these compounds to tuna quality and other aspects, as it was done in lines 154-155 for HxR and Hx. Please add a reference after this sentence (line 155).
Lines 167-177: Authors discuss about the possible origins of the identified VOCs, finally. However, this should be done in a more concise and focused manner: please extract the main changes and connect them to the possible causes, supported by relevant literature citations.
Lines 182-183: There’s no need to repeat the data from the table in details.
Line 186: “...a fishy odour was confirmed...” – there is no reference. Did the authors perform sensory analysis? If so, please include details in the Materials and methods section.
Lines 187-189: Authors stated that aldehydes could be looked upon as indicators of the degree of the oxidation of polyunsaturated fatty acids which can be used as a criterion for the evaluation of freshness. In my opinion, such statements could also find their place in the introduction section, but have to be supported by references.
Line 195: “…as shown in an earlier study (30)”.
Line 198: “...was previously related…”
The whole Results and discussion section: please be more focused in extracting the most relevant changes during storage clearly and concisely, relate these to previous findings, and try to explain what happened based on the current knowledge. Use appropriate and contemporary references. Please emphasize more the role of water and its connection with the development of VOCs; it is mentioned the first and only time by the end of the manuscript (line 244-257), should be mentioned and clearer early in the text.
Round 2
Reviewer 3 Report
It is in my opinion that now the manuscript is significantly improved and deserves publication in Molecules after the correction of certain minor details, and a minor improvement of English language at some points.
Line 197: Please include a reference on the possible negative effects of histamine.
Line 214: Authors decided not to perform 2-way ANOVA which can be tolerated. However, when comparing 0 C and 4 C, please do not use the phrases like "dramatically" and similar, since 1-way ANOVA did not reveal any significant differences between these temperatures at any point.
